# Short-horizon neonatal seizure prediction using EEG-based deep learning

Jonathan Kim[1], Edilberto Amorim[2], Vikram R. Rao[2], Hannah C. Glass[2,3,4], Danilo Bernardo[2]*

**1** Department of Neurology and Neurologic Sciences, Stanford University. Palo Alto, California, United States of America, **2** Department of Neurology and Weill Institute for Neurosciences, University of California San Francisco, San Francisco, California, United States of America, **3** Department of Epidemiology and Biostatistics, University of California San Francisco, San Francisco, California, United States of America, **4** Department of Pediatrics, University of California San Francisco, San Francisco, California, United States of America

* dbernardoj@gmail.com

## Abstract

Strategies to predict neonatal seizure risk have typically focused on long-term static predictions with prediction horizons spanning days during the acute postnatal period. Higher temporal resolution or short-horizon neonatal seizure prediction, on the time-frame of minutes, remains unexplored. Here, we investigated quantitative electroencephalography (QEEG) based deep learning (DL) for short-horizon seizure prediction. We used two publicly available EEG seizure datasets with a total of 132 neonates containing a total of 281 hours of EEG data. We benchmarked current state-of-the-art time-series DL methods for seizure prediction, identifying convolutional LSTM (ConvLSTM) as having the strongest performance at preictal state classification. We assessed ConvLSTM performance in a seizure alarm system over varying short-range (1–7 minutes) seizure prediction horizons (SPH) and seizure occurrence periods (SOP) and identified optimal performance at SPH 3 min and SOP 7 min, with AUROC 0.8. At 80% sensitivity, false detection rate was 0.68 events/hour with time-in-warning of 0.36. Model calibration was moderate, with an expected calibration error of 0.106. These findings establish the feasibility of short-horizon neonatal seizure prediction and warrant the need for further validation.

## Author summary

Neonatal seizures are associated with substantial long-term morbidity and mortality. A promising strategy to improve neonatal clinical outcomes has focused on developing EEG-based machine learning models to identify seizure-prone neonates to reduce the time to seizure diagnosis and treatment. Such work has typically focused on identifying neonates at risk of seizure, providing static predictions, rather than dynamic predictions containing information of when the

**Data availability statement:** The data from Helsinki University Hospital that support the findings of this study are publicly available from https://zenodo.org/records/4940267 with DOI 10.1038/sdata.2019.39. The data from Cork University Hospital that support the findings of this study are publicly available from https://zenodo.org/records/7477575 with DOI: 10.1038/s41597-023-02002-8.

**Funding:** The author(s) received no specific funding for this work.

**Competing interests:** The authors have declared that no competing interests exist.

seizure may occur. Here, we focus on short-term seizure prediction to advance the temporal resolution of neonatal seizure prediction. We present a deep learning approach that leverages quantitative EEG to predict seizures on the order of minutes. We demonstrate that short-term seizure prediction—down to a 7-minute horizon—is accurate. Our results suggest that precise real-time estimation of dynamic seizure risk is feasible, potentially enabling improved triage of resources to neonates at higher risk for seizures and earlier clinical intervention for neonatal seizures. However, further development and external validation are warranted.

## Introduction

Neonatal seizures, with an incidence rate of one to three per 1000 life births, are associated with substantial long-term morbidity and mortality [1,2]. Prompt seizure treatment is crucial, as a higher seizure burden is associated with increased treatment resistance and mortality [3–6]. A promising strategy to improve neonatal clinical outcomes has focused on identifying at-risk neonates thereby reducing the time to seizure diagnosis and treatment [7–11]. Real-time seizure prediction may enable alerts of impending seizure onset (Fig 1), which may be used to guide acute therapeutic strategies, such as prophylactic interventions in higher seizure risk populations such as in neonatal encephalopathy NE [12–14]. In addition, seizure prediction in neonates has the potential to improve neonatal care by optimizing the allocation of EEG monitoring resources, particularly in environments with limited

### Short-term vs long-term seizure risk prediction

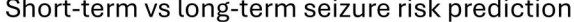
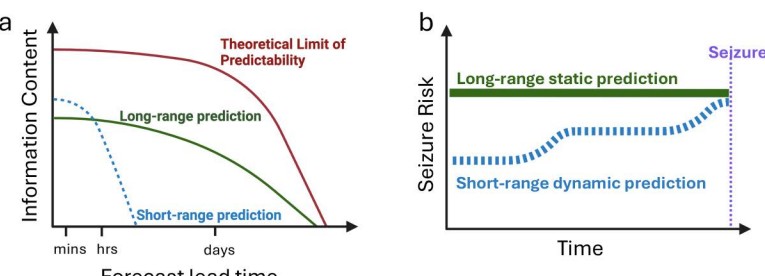

**Fig 1. Comparison of short- versus long-horizon predictions.** (A) illustrates differences between short- and long-horizon predictions. Short-range or short-horizon predictions may be updated frequently (indicated by dynamics in the dashed line), which enables these predictions to be revised regularly as new data becomes available. Current neonatal long-range seizure prediction typically provide time-invariant, or static predictions. (B) illustrates the relation between information content and prediction lead time, following analogous prediction definitions previously developed in meteorology [49]. Here, *information content* may pertain to the degree of accuracy, certainty, and overall applicability of the predictions. There is a downward trend in information content in the theoretical limit of predictability, short- and long-range prediction curves reflecting inevitable increase in uncertainty as one attempts to predict further into the future with increasing lead time, or SPH. The short-horizon prediction may potentially contain higher information content than long-range predictions during short lead times because it is based on more up to date observations.

EEG availability [15], such as in low-income countries, where only 25% centers reported having any access to EEG within their centers [16].

Recent studies have leveraged machine learning (ML) to predict seizures in neonatal encephalopathy (NE) with high accuracy utilizing long prediction horizons [10,11]. Pavel et al. introduced a multimodal ML model, utilizing clinical variables and quantitative EEG (QEEG) features shortly after birth, to predict neonates with NE who developed seizures, predicting individual seizure risk over several days [10]. Similarly, McKee et al. developed a ML model that utilizes qualitative EEG with a seizure prediction horizon of several days [11]. More recently, Bernardo et al. demonstrated feasibility of utilizing ML with clinical and QEEG variables to estimate time-varying seizure risk with prediction horizon of several hours [17]. While these and other prior studies have predicted seizure risk over an observation period spanning several hours to days [7–11], high temporal resolution or short-horizon seizure prediction remains unexplored in the neonatal population. This contrasts with the adult and pediatric populations, where ML studies have demonstrated feasibility of predicting seizures from 2 to 50 minutes before onset [18–20].

Compared to long-horizon static predictions, dynamically updated short-horizon predictions may contain higher information content in the short-term because they are based on more up to date observations (Fig 1A). Thus, short-horizon predictions may provide increased temporal precision of when seizures may occur (Fig 1B) potentially improving allocation of monitoring resources and enabling acute therapeutic strategies, such as administration of prophylactic antiseizure therapy [21]. Our objective was to develop a deep learning (DL) approach for neonatal seizure prediction, with a focus on the short-range prediction horizon.

## Materials and methods

### Subject data

We analyzed publicly available neonatal EEG datasets: 1) Helsinki University Hospital (HUH) dataset consists of EEG recorded from 79 term neonates, with total duration of 111.9 hours, with seizures annotated by three experts [22] and 2) the INFANT Research Center, Cork University Maternity Hospital, (Cork) dataset consists of EEG records from 53 neonates affected by HIE with total duration 169 hours [23] Whereas the majority of HUH subjects (52%) contained seizures, only two Cork subjects only two Cork subjects had seizures, thus providing more balance between subjects with versus without seizure. We note that seizure etiologies in the HUH dataset included acute events such as encephalitis/meningitis, cerebral infarction, HIE, and intracranial hemorrhage. Together, these comprise acute etiologies which constitute the majority (approximately 75%) of neonatal seizure cases, indicating that our dataset is broadly representative of the neonatal seizure population [24].

### Ethics statement

As we utilized the publicly available HUH and Cork EEG datasets, we reference their reported consent and Institutional Review Board or Ethics Committee as follows. For the HUH dataset, data was acquired as standard of care at the Helsinki University Hospital, Finland and permission to publicly release de-identified EEG was obtained from the local ethics committee of the Helsinki University Hospital Ethics Committee as described by Stevenson et al [25]. For the Cork dataset, written and informed consent was obtained from the neonate's guardian or parent, and permission to record and publicly release de-identified EEG was obtained from the Data Protection Officer at University College Cork, Ireland, as described by O'Toole et al [23].

### Preprocessing

We preprocessed and segmented EEG data into preictal (6–1 min before seizures), interictal (1 min post-seizure to 5 min before next), and ictal epochs (annotated by ≥2 experts), following Bernardo et al [17]. EEG data was band-pass filtered

between 0.1 Hz and 20 Hz, as electromyogenic and electrical artifacts limits the signal to noise ratio of EEG acquisition above 20 Hz in neonatal ICU contexts, then resampled to 40 Hz. The selected preictal window immediately before a seizure is hypothesized to facilitate seizure onset, and has been previously used for preictal state classification [26,27].

Right-censored epochs occurring at the end of data recordings, where seizure occurrence within the prediction horizon is unknown, were excluded. To improve model generalizability, we augmented the training data using translation-invariant transforms by transposing EEG channels along frontal-occipital and left-right axes and recomputing training features.

## Feature selection

We previously assessed and ranked a broad set of QEEG features highly predictive of seizure onset using the Boruta feature selection algorithm [28,29], and here utilize the top three feature categories: standard summary statistics (mean, standard deviation, kurtosis, skew, the 10th percentile, and the 90th percentile). We selected only three feature families in order to limit the number of covariates included in the model to reduce the risk of overfitting given our sample size [30]. The predictive performance of these features have been validated in an external HIE dataset [17]. Features were calculated across each montaged channel left-right pair, power spectral features calculated at each montage channel [31], and recurrence quantification analysis features calculated at each montage channel [32]. Further details of QEEG features calculated are detailed in S1 Table and the code used to generate QEEG features is available at https://github.com/dbernardo05/qeegfeats. In addition, code used to train DL models is available at https://github.com/dbernardo05/neoSeer. We acknowledge that several of the features listed were originally shared by participants in the Kaggle Seizure Prediction Contest [33].

## Model training

We evaluated current state-of-the-art DL-architectures for time-series, utilizing time-varying QEEG features for seizure prediction. We evaluated ConvLSTM, ResNet, Transformer, Time-series Transformer, InceptionTime, and OmniscaleCNN time-series DL models. For baseline comparison, we also evaluated ML models including: Support Vector Machine, K-Nearest Neighbors, Logistic Regression, and Random Forest classifiers. Details regarding implementation of ML and DL models are found in S1 and S2 Methods, respectively. The pytorch and pytorch-lightning libraries were used to evaluate DL models [34]. The model was trained using binary cross-entropy loss to penalize misclassification between preictal and interictal classes. We performed nested 10-fold cross-validation, splitting data per fold into training (67.5%), validation (22.5%), and test (10%) sets, with inter-subject stratification, to prevent data leakage.

## Seizure prediction framework and evaluation

We first evaluated the performance of ML and DL models at classification of preictal versus interictal state. Details regarding performance metrics are discussed in S3 Methods. Preictal state classification models can be utilized as part of a seizure prediction alarm system [35]. To evaluate predictions in a simulated clinical context, we employed a commonly used seizure alarm framework, which utilizes the seizure prediction horizon (SPH) and seizure occurrence period (SOP) [36]. A minimum SPH is designated to ensure sufficient lead time preceding a seizure to allow for timely intervention strategies to be employed. From a clinical perspective it is advantageous to have a longer SPH and a shorter SOP – in other words, to have more time in which to intervene within the SPH, and to have less uncertainty regarding time of seizure onset during the SOP. In our evaluation, we systematically vary the SOP and SPH to find an optimal set of values for the general use case. The combined duration of SOP and SPH is equivalent to the alarm warning period. For seizures that are less than the SPH from the previous seizure, we consider them as a single seizure event. Further details regarding the definitions of SPH and SOP are shown in S1 Fig.

In this system, a seizure warning is issued when the probability of a preictal state surpasses a predefined decision threshold. In the performance assessment, this decision threshold is systematically adjusted in order to derive confusion

matrix metrics at each threshold setting. The Area Under the Receiver Operating Characteristic (AUROC) was subsequently calculated using the true positive rate (TPR) and false positive rate (FPR) across these varying thresholds. We computed Sensitivity as defined by Proix et al. as the number of time points in alarms with correctly predicted seizures divided by total number of time points in alarm states with seizures [37]. The calculated corrected proportion of time in warning (TIW) was calculated as (TIW – time in true alarm states)/ (total time – time in true alarm states). Additionally, we calculated Area under the Precision-Recall Curve (AUPRC), which is particularly informative in imbalanced datasets such as in this case, where interictal periods (negative class) significantly exceed preictal periods (positive class). For AUPRC, we computed area under the curve of Recall (equivalent to the previously defined Sensitivity) versus Precision which was defined as equal to the total number of time points in alarms with correctly predicted seizures divided by total number of time points in all alarm states (including both correct and incorrect predictions).

To evaluate the calibration and predictive skill of the seizure prediction system, we employed Expected Calibration Error (ECE) and Brier Skill Score (BSS). ECE is evaluated to measure the reliability of the neural network model's predicted probabilities and is commonly used to evaluate neural network predictive performance [38]. It assesses the discrepancy between the predicted and true probabilities of the outcomes, which evaluates the calibration of the model. The BSS assesses the model's skill relative to a reference random prediction model. The BSS considers both calibration and discrimination of the model. A positive BSS indicates that the model performs better than the reference model, while a negative BSS signifies the opposite. For the BSS calculation, we utilized the standard climatology reference, which accounts for the prevalence of the positive class (preictal states).

## Results

### Study subjects

The Cork EEG dataset comprises 53 subjects with median age 39.5 (interquartile range: 37.8-40.5) weeks, of which two subjects (4%) experienced seizures [23]. The HUH EEG dataset comprises 79 subjects with median age 40 weeks (39.4 - 40.7), among whom 39 subjects (49%) experienced seizures, totaling 516 seizures [25]. A total of 281 hours of EEG data were recorded across all subjects. The median EEG recording durations per subject of the HUH and Cork datasets were 1.2 (1.1-1.6) hrs and 3 (2–4) hrs, respectively. The median seizure burden across all subjects who experienced seizures was 5 (2–9) seizures, with median seizure duration 1.23 (0.66-2.7) mins. Additional clinical and EEG details regarding these datasets is reported in S2 Table and details regarding dataset size per cross-validation fold is reported in S3 Table.

### Seizure risk prediction

Examples of the resulting time-varying seizure risk predictions for individual neonates with seizures are shown in Fig 2. In these examples, periods containing heightened seizure risk occur with varying lead times prior to the seizure occurrences (red lines). In comparison, the selected examples from neonates with no seizures show no peaks indicative of preictal state (Fig 3). To select the best performing model for seizure alarm evaluation, we first benchmarked model ability to distinguish preictal from interictal states, a prerequisite for seizure prediction [35]. We evaluated current state-of-the-art DL methods by comparing performance using QEEG features (Table 1) and found that convolutional-LSTM utilizing QEEG (ConvLSTM-QEEG) features outperformed other DL methods, including those utilizing automated feature extraction (S4 Table), and conventional ML methods (S5 Table), with AUROC of 0.742 (standard error 0.103) and AUPRC of 0.241 (0.130). Thus, we selected the ConvLSTM architecture for subsequent seizure alarm system evaluation. Given the different prevalence of seizures in the subgroups, we performed a subgroup analysis comparing balanced accuracy for the HUH vs Cork datasets (S5 Fig). This analysis demonstrated that while preictal classification accuracy exceeded chance levels (50%) for most subjects, the performance was relatively higher in the Cork dataset compared to the HUH dataset.

# Subjects with seizures

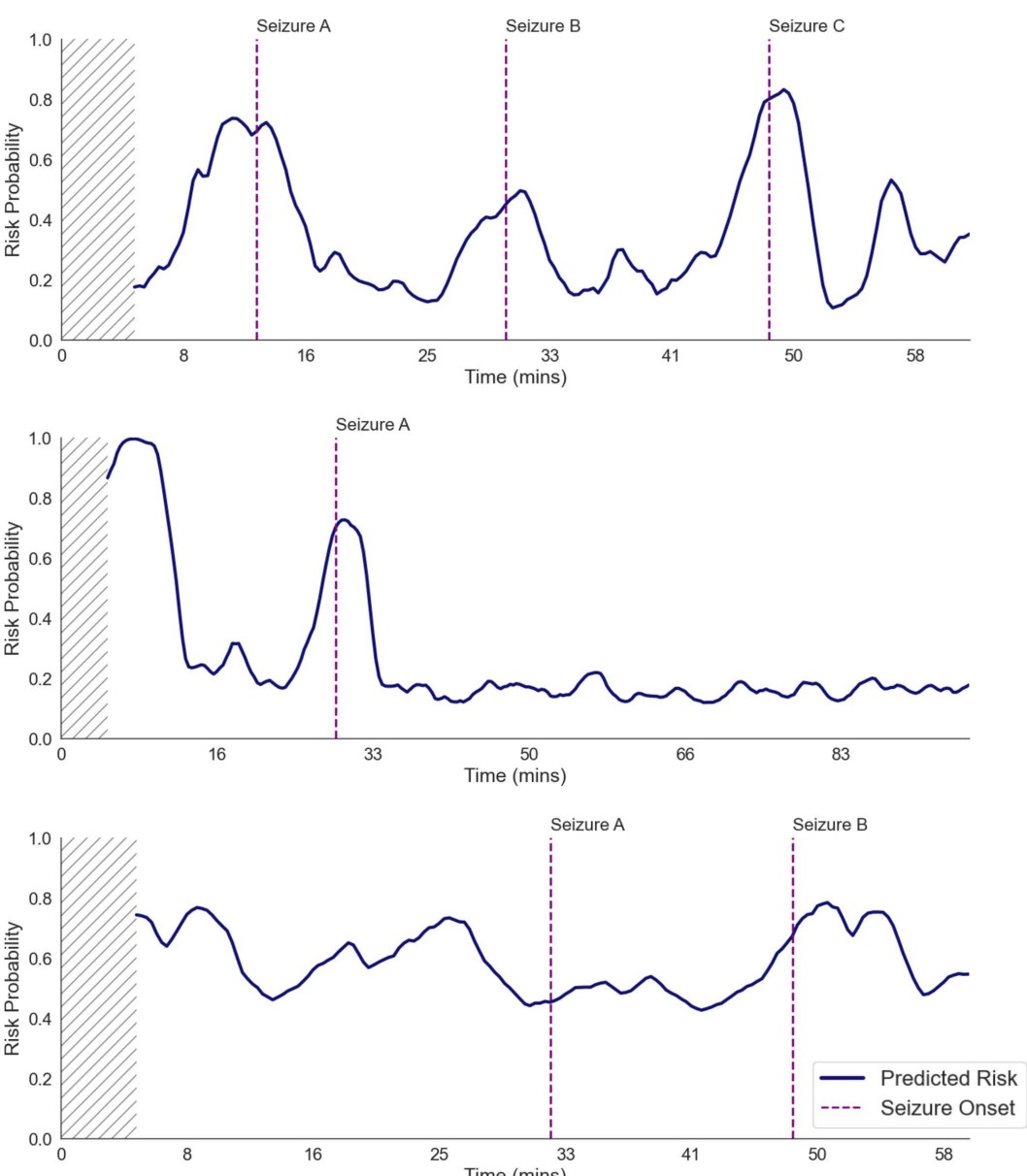

**Fig 2. Examples of dynamic seizure prediction risk probabilities in subjects with seizure.** Dynamic seizure prediction risk probabilities from 3 subjects with seizures are shown. Prior to seizure occurrences, there are increases in estimated preictal state probability. There are also elevations in preictal state probability not associated with immediate seizure. The hashed region indicates the model's burn-in period, during which predictions are unavailable due to insufficient length of data.

# Subjects without seizures

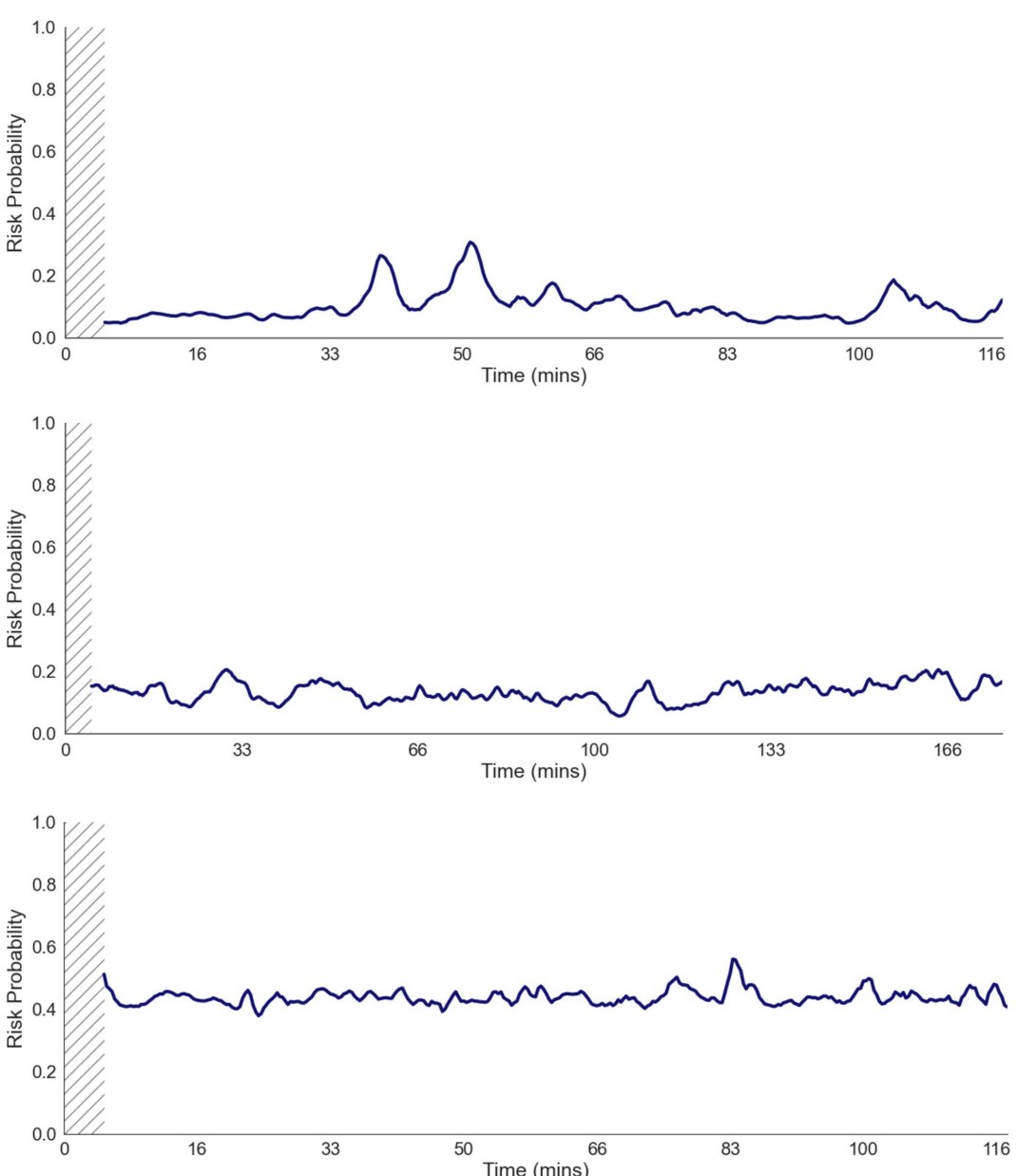

**Fig 3. Examples of dynamic seizure prediction risk probabilities in subjects without seizure.** Dynamic seizure prediction risk probabilities from 3 subjects without seizures are shown. There are no significant elevations in preictal state probability seen. The hashed region indicates the model's burn-in period, during which predictions are unavailable due to insufficient length of data.

**Table 1. Comparison of DL methods using QEEG.**

| Model | AUROC* | AUPRC | F1 |
|---|---|---|---|
| ConvLSTM | **0.742 (0.103)** | **0.241 (0.130)** | 0.453 (0.092) |
| InceptionTime | 0.738 (0.103) | 0.239 (0.143) | 0.200 (0.079) |
| OmniScaleCNN | 0.705 (0.107) | 0.237 (0.178) | **0.492 (0.062)** |
| ResNeT | 0.707 (0.090) | 0.231 (0.182) | 0.252 (0.128) |
| TCN | 0.675 (0.068) | 0.202 (0.107) | 0.252 (0.131) |
| TSiT | 0.676 (0.134) | 0.214 (0.091) | 0.303 (0.083) |
| Transformer | 0.713 (0.081) | 0.235 (0.133) | 0.341 (0.103) |

† The ConvLSTM AUROC was compared to AUROC for other models using the Hanley McNeil test, and the differences in the AUROC were statistically significant between ConvLSTM and other architectures. ConvLSTM generally outperformed other DL methods utilizing QEEG. Data reported as average performance across all cross-validation folds (10) with (standard error). The baseline AUPRC was 0.110. Details regarding calculation of AUROC, AUPRC, and F1 are discussed in S3 Methods. Abbreviations: Area Under the Receiver Operator Characteristic (AUROC), Area Under the Precision Recall Curve (AUPRC), F1 score (F1), Time Series Transformer (TSiT, based on vision transformer, VIT), Temporal convolutional network (TCN).

## Seizure alarm system evaluation

To assess performance in a simulated clinical setting, we adopted a widely used seizure alarm framework that incorporates the seizure prediction horizon (SPH) and seizure occurrence period (SOP) [36]. Under this system, a seizure alert is triggered when the probability of a preictal state surpasses a predefined decision threshold [35]. To identify optimal SPH and SOP for short-range prediction horizon, we evaluated AUROC and AUPRC under varying SOP and SPH between 1–7 minutes (Fig 4). ConvLSTM-QEEG had peak AUROC of 0.80 under SPH of 3 minutes and SOP of 7 minute. At these SPH and SOP parameters, with ROC threshold adjusted to correspond with 80% sensitivity, there was a corresponding false detection rate (FDR) of 0.68 events/hr (S2a Fig). S2b Fig demonstrates that FDR were unevenly distributed, with certain subjects experiencing disproportionately higher FDR. These tended to be subjects without observed seizures, who may nevertheless represent neonates with true risk of seizure despite not having seizures during the recorded period. The peak AUPRC was 0.23 with SPH of 1 minute and SOP of 7 minutes, with corresponding baseline AUPRC 0.0450. For any given SPH, increasing the SOP led to improved AUROC and AUPRC, reflective of increased ease of prediction at lower temporal resolutions. However, this resulted in increased percentage of time in warning (TIW), with TIW ranging from 0.33 to 0.37 (S3 Fig). At the subject-level, there did not appear to be any apparent clustering of cases where the model performed extremely well or poorly (S4 Fig).

Next, we examined the impact of varying prediction horizon on the Brier Skill Score (BSS), which quantifies the difference in predictive performance between our model and a reference model. We used the baseline prevalence as the reference comparator. The effect of varying SPH and SOP on BSS is shown in Fig 5B. The highest BSS was 0.056 obtained with a SPH of 1 minute and SOP of 10 minutes. We find that for all SOP, as the SPH increases, then model performance concomitantly decreases. This follows the intuition that prediction farther into the future is inherently more difficult. Similarly, as the SOP temporal resolution becomes finer, model performance also decreases, suggesting that predicting with increased temporal resolution is also more difficult.

## Model calibration

The calibration of the ConvLSTM model was evaluated using the reliability plot and Expected Calibration Error (ECE) metrics. The reliability plot, (Fig 5A), demonstrates the relationship between the predicted probabilities and the observed frequency of the preictal class. There is a clustering of points below the diagonal, which signifies that ConvLSTM was occasionally overconfident in its predictions for the preictal class at these probabilities. Additionally, isolated points distant from the diagonal occurred at the lowest and highest probability bins, consistent with respective isolated underconfident

### a  AUROC Scores under varying SOP and SPH

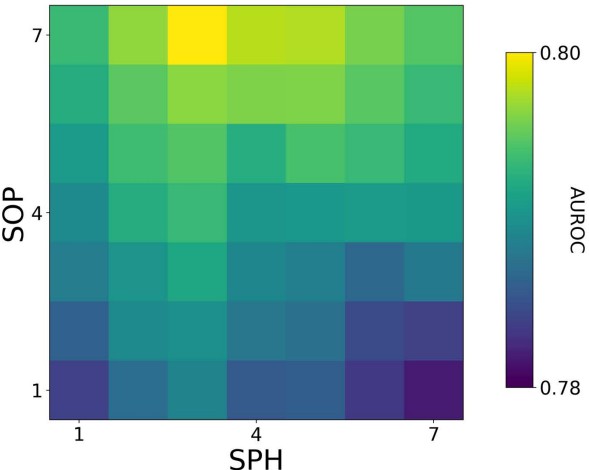

### b  AUPRC Scores under varying SOP and SPH

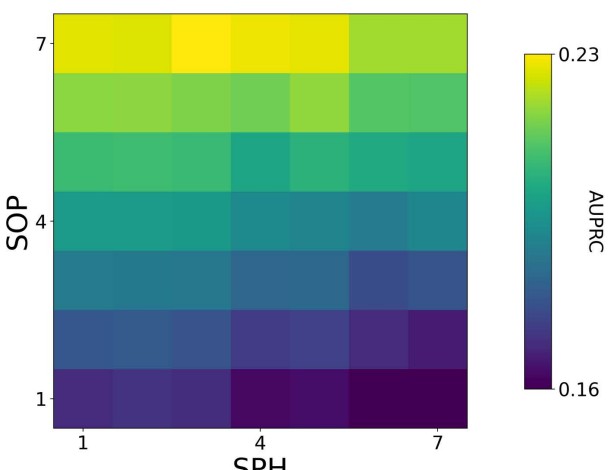

**Fig 4. Area under the ROC and PR curve analyses.** (A) shows respective ConvLSTM AUROC's for SOP and SPH varied between 1 and 7 minutes. The best AUROC was 0.80 at SOP of 7 minutes and SPH of 3 minutes. (B) shows respective AUPRC's for SOP and SPH varied between 1 and 7 minutes. The best AUPRC was 0.23 at SOP of 7 minutes and SPH of 3 minutes.

and overconfident predictions at these bins. Concordant to the reliability plot findings, the ECE value of 0.106 is consistent with a moderately well-calibrated model. The value of 0.106 is slightly above the general desirable range for a strongly calibrated model, considered less than 0.1 (zero indicates perfect calibration, and one is the maximum value indicative of weak calibration).

## Discussion

In this study, we demonstrate the feasibility of short-term, real-time prediction of neonatal seizures. In contrast to prior neonatal seizure prediction studies, which estimate neonatal seizure risk over several days to guide extended EEG monitoring and prognostication [10,11], our approach offers minute-scale, dynamic predictions that complement these

**a**    Reliability Plot and Expected Calibration Error ($\varepsilon$)

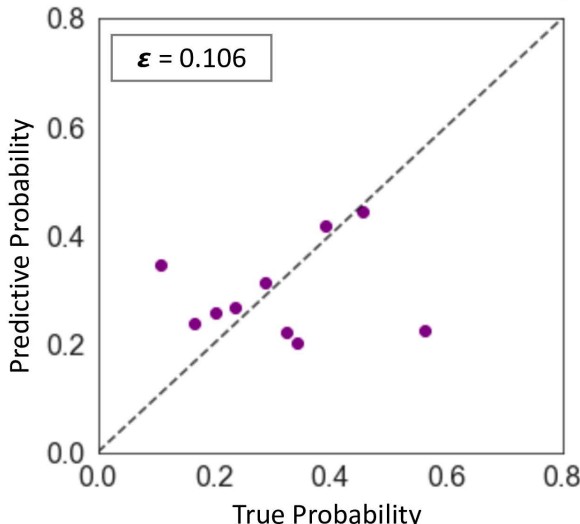

**b**    Brier Skill Score dependence on SPH and SOP

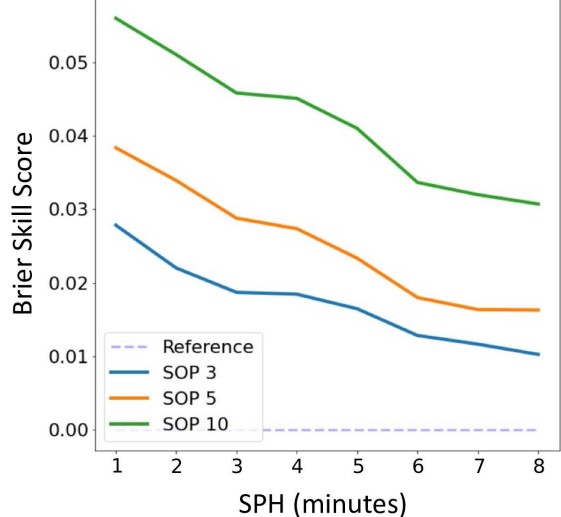

**Fig 5. Reliability plot and Brier Skill Score.** (A) demonstrates ConvLSTM calibration evaluation using a reliability plot and Expected Calibration Error (ECE) metric. In the ECE plot, the ECE ($\varepsilon$) value of 0.106 indicates a moderately-well calibrated model. (B) demonstrates the influence of varying SPH and SOP on the Brier Skill Score (BSS). The best BSS occurred with SPH of 1 minute and SOP of 10 minutes. An increase in SPH correlates with a decline in BSS, and analogously, as the SOP decreases, model performance also decreases, indicating that prediction with increasing lead-times and increasing temporal resolution becomes more challenging.

longer-range assessments. By providing more precise, dynamic updates on seizure risk, these short-term predictions may facilitate time-sensitive interventions, such as administration of rapid-acting antiseizure prophylaxis, enable real-time adjustments to resource allocation such as more intensive monitoring. Specifically, we utilized a short preictal period to focus on dynamical EEG changes that herald seizures, an approach for seizure prediction first investigated by Ghademi et al [26,27]. To our knowledge, this is the first study to demonstrate short-term neonatal seizure prediction.

Our AUROC result of 0.8 was comparable to performance of a generalized seizure prediction model in patients with chronic electrocorticography recordings which reported AUROC between 0.68 to 0.70 [39]. Compared to studies which have utilized subject-specific models (S6 Table), we report relatively higher false alarm rate and lower sensitivity. We attribute this to our subject-independent approach which is expected to have poorer aggregate performance than subject-specific model approaches. Indeed, on a per-subject analysis in subjects containing seizures, at least one-third of the subjects had AUROC < 0.5 (S4 Fig), indicating that the model poorly generalizes for certain subjects. In addition, the relatively lower performance seen in this study may reflect the usage of short horizon SOP and SPH, as predicting with increasingly higher temporal resolution is considered more difficult in complex systems, such as in seismology or meteorology, and this has also been suggested for seizure prediction [33,40,41]. Although our performance metrics are moderately lower than those attained in prior patient-specific models in pediatrics, the subject-independent approach obviates the need for subject-specific training, which is advantageous for neonates who are at immediate elevated risk of seizure after birth when subject-specific data is not be immediately available to train a prediction model before the first seizure. Furthermore, subject-independent models facilitate efficient resource allocation, as it can be readily implemented at different sites without fine-tuning, reducing the need for specialized expertise and computational resources. In addition, our dynamic approach facilitates real-time risk updating, as predictions are updated at 20-second rolling intervals, laying the foundation for adaptation to continuous EEG data streams in clinical settings.

The use of DL models for seizure prediction has previously been investigated in the adult and pediatric populations [18–20]. To the best of our knowledge, our findings are the first demonstration of utilizing DL for neonatal seizure prediction. We found that the strongest performing DL architecture to be convolutional long short-term memory neural network (ConvLSTM), an architecture that has previously been utilized for seizure prediction [42]. ConvLSTM incorporates strengths of both convolutional neural network (CNN) and LSTM architectures: the convolutional module extracts short-term temporal features predictive of seizure risk, while the LSTM module learns long-range temporal dependencies. Prior studies in seizure prediction have previously utilized CNN [18,20,43], LSTM [19,44], and CNN-LSTM [42]. In contrast to the previously published CNN-LSTM method, which utilized CNN-LSTM on short-time Fourier-transformed (STFT) EEG signal, we incorporated QEEG features.

There are several limitations of our study. Primarily, we acknowledge that we did not utilize a held-out or independent evaluation dataset. Thus, future validation on larger and independent datasets is warranted.In addition, potential bias may arise from our selection of a 5 minute preictal window. While this choice aligns with findings identifying an ideal preictal period of 10 minutes as optimal for preictal versus interictal classification in the pediatric population [45], it potentially loses information by overlooking more gradual, longer evolving QEEG features indicative of the pro-ictal state [46]. This limitation potentially reduces the prediction horizon. Furthermore, we acknowledge that predictive probabilities from DL may be poorly calibrated in the task of time-series prediction [47]. Indeed, we demonstrated that ConvLSTM probabilities were only moderately well-calibrated, suggesting potential for providing poorly calibrated predictions, necessitating future work for further validation and calibration improvement before being utilized in practice. Lastly, although real-time neonatal seizure prediction has the potential to improve resource allocation in resource-limited countries, we acknowledge that technological availability alone is insufficient to translate into improved clinical practice. As highlighted by Nji et al. regarding AlphaFold's potential to advance biological research in low-resource settings, concurrent development in research infrastructure and science policies are essential for disseminating technological innovations in these contexts [48].

In conclusion, we have demonstrated the feasibility of applying DL to facilitate short-horizon neonatal seizure prediction. Future validation and refinement is warranted, especially with respect to predictive accuracy and calibration. Short-term seizure prediction in neonates may facilitate time-sensitive interventions in neonates at high seizure risk and potentially improve the triage of EEG monitoring resources, particularly in low-resource settings.

## Supporting information

**S1 Fig. Seizure alarm system.** The seizure prediction horizon (SPH) and seizure occurrence period (SOP) evaluation framework as proposed by Maiwald et al. considers that there should be a minimum SPH to provide ample lead time before a seizure to allow for intervention and that the alarm should have SOP selected to align prediction duration with the specified clinical observation period. Following Maiwald et al., *"the SOP is defined as a time period during which the seizure is to be expected"* and the SPH *"is a minimum window of time between the alarm raised by the prediction method and the beginning of SOP"* (41). The system triggers an alarm, lasting the combined duration of SPH and SOP, if the designated seizure threshold is met. At time $t$, a true positive alarm occurs if a seizure initiates between $t+$SPH and $t+$SPH$+$SOP; otherwise, a false positive is marked. A false negative occurs if a seizure occurs at time $t_s$ and no alarm is activated. A true negative occurs when no alarm is triggered and no seizure occurs.
(TIF)

**S2 Fig. FDR versus Sensitivity. A.** The relation between FDR and sensitivity is shown for ConvLSTM in the seizure prediction task with SPH of 3 minutes and SOP of 7 minutes. There is a resultant increase in FDR as sensitivity increases. The operating point corresponding with 80% sensitivity is denoted by the orange X, and this corresponds with an FDR of approximately 0.7. Details regarding sensitivity and FDR calculation are discussed in Methods, Seizure Prediction System Design and Evaluation Section, Paragraph 3. **B.** FDR was unevenly distributed across subjects, with certain subjects experiencing disproportionately higher FDR, particularly those without observed seizures.
(TIFF)

**S3 Fig. AUROC dependence on Time in Warning and SOP.** The relation between AUROC with Time in Warning (TIW) and Seizure Occurrence Period (SOP) is shown for ConvLSTM in the seizure prediction task with SPH of 3 minutes and varying SOP between 1–7 minutes. There is a resultant increase in AUROC and Time in Warning as SOP increases.
(TIFF)

**S4 Fig. AUROC and AUPRC scores for all subjects.** Individual AUROC and AUPRC scores are demonstrated on a per subject basis for all subjects who had seizures. There is no apparent clustering of subjects, and in general, performance AUROC and AUPRC appear correlated. Of note, only subjects who contained seizures were included in this analysis because AUROC calculation requires seizure occurrences for calculation of true and false positive rates.
(TIFF)

**S5 Fig. Subgroup analysis comparing balanced accuracy for the HUH and Cork.** Preictal classification accuracy exceeded chance levels (50%) for most subjects, however, the performance was relatively higher in the Cork dataset compared to the HUH dataset.
(TIFF)

**S1 Table. Features belong to three feature families (left column) including power spectral features, asymmetry indices of statistical moments, and recurrence quantification analysis were calculated and parameterized as described in the right column.** Asymmetry indices features were calculated across each montaged channel left-right pair, power spectral features calculated at each montage channel, and recurrence quantification analysis features calculated at each montage channel.
(DOCX)

**S2 Table. EEG dataset characteristics from the Cork and HUH datasets are shown, including indications for monitoring, clinical characteristics, and EEG characteristics.**
(DOCX)

**S3 Table. K-fold characteristics of each stratified K-fold split, including 20 second epochs per fold and hours per fold, are demonstrated.**
(DOCX)

**S4 Table. Comparison of different deep learning (DL) methods using automated feature extraction (AFE) is demonstrated.** The ConvLSTM-QEEG AUROC was compared to AUROC for other models using the Hanley McNeil test, and the differences in the AUROC were statistically significant between ConvLSTM and other architectures. In the task of preictal versus interictal state classification, ConvLSTM-QEEG (Table 1) generally outperformed DL methods utilizing automated feature extraction (AFE) directly on EEG, as presented here in Supplementary Table 4. Notably, certain DL models with AFE outperformed in metrics in isolation, with ConvLSTM-AFE obtaining higher AUROC (0.716), however, ConvLSTM-QEEG had the best performance across all metrics. It should be noted that some of the more recent time-series DL models (e.g., TSiT, OmniScaleCNN, and InceptionTime) were developed and validated on the UC Riverside Time-Series Classification dataset which, while expansive with some biological data including heartbeat and atrial fibrillation time-series, does not contain EEG data. Data reported as average performance across all cross-validation folds (10) with (standard error). Details regarding calculation of AUROC, AUPRC, and F1 are discussed in Supplementary Methods 3.
(DOCX)

**S5 Table. Comparison of conventional ML methods.** We benchmarked conventional ML methods which all utilized QEEG features and demonstrate that DL approaches generally demonstrated improved performance compared to ML methods. Data is reported as average performance across all cross-validation folds (10) with (standard error). Details regarding calculation of AUROC, AUPRC, MCC, and F1 are discussed in Supplementary Methods 3.
(DOCX)

**S6 Table. Summarization of prior seizure prediction studies that have utilized subject-specific models, including parameterization of seizure prediction horizon and seizure occurrence period, with resultant predictive performance including sensitivities and false positive rates.**
(DOCX)

**S1 Methods. Machine learning (ML) methodology used for ML model comparators is described, including training/optimization schema and parameterization.**
(DOCX)

**S2 Methods. Deep learning (DL) methodology used for DL models including ConvLSTM, ResNet, Transformer, Time-series Transformer, InceptionTime, and OmniscaleCNN is described, including including training/optimization schema and parameterization.**
(DOCX)

**S3 Methods. Classification model performance evaluation methodology including calculation of performance metrics (AUROC, AUPRC, F1, expected calibration error and others) is described.**
(DOCX)

## Acknowledgments

We are deeply appreciative to the many participants in the Kaggle Seizure Prediction Contest who have graciously shared their quantitative EEG algorithms, which we have utilized in this work [33].

## Author contributions

**Conceptualization:** Jonathan Kim, Edilberto Amorim, Vikram R. Rao, Hannah C. Glass, Danilo Bernardo.

**Data curation:** Jonathan Kim.

**Methodology:** Danilo Bernardo.

**Software:** Jonathan Kim.

**Supervision:** Danilo Bernardo.

**Writing – original draft:** Jonathan Kim, Edilberto Amorim.

**Writing – review & editing:** Jonathan Kim, Edilberto Amorim, Vikram R. Rao, Hannah C. Glass, Danilo Bernardo.

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
