## [Decision Letter · Decision Letter 0]

PDIG-D-25-00031Short-Horizon Neonatal Seizure Prediction Using EEG-Based Deep LearningPLOS Digital Health Dear Dr. Bernardo, Thank you for submitting your manuscript to PLOS Digital Health. After careful consideration, we feel that it has merit but does not fully meet PLOS Digital Health's publication criteria as it currently stands. Therefore, we invite you to submit a revised version of the manuscript that addresses the points raised during the review process. Please submit your revised manuscript within 60 days Apr 20 2025 11:59PM. If you will need more time than this to complete your revisions, please reply to this message or contact the journal office at digitalhealth@plos.org. Please include the following items when submitting your revised manuscript:* A rebuttal letter that responds to each point raised by the editor and reviewer(s). You should upload this letter as a separate file labeled 'Response to Reviewers '. This file does not need to include responses to any formatting updates and technical items listed in the 'Journal Requirements' section below.* A marked-up copy of your manuscript that highlights changes made to the original version. You should upload this as a separate file labeled 'Revised Manuscript with Track Changes '.* An unmarked version of your revised paper without tracked changes. You should upload this as a separate file labeled 'Manuscript '. If you would like to make changes to your financial disclosure, competing interests statement, or data availability statement, please make these updates within the submission form at the time of resubmission. Guidelines for resubmitting your figure files are available below the reviewer comments at the end of this letter. We look forward to receiving your revised manuscript. Kind regards, Hualou LiangAcademic EditorPLOS Digital Health Hualou LiangAcademic EditorPLOS Digital Health Leo Anthony CeliEditor-in-ChiefPLOS Digital Healthorcid.org/0000-0001-6712-6626 **Journal Requirements:**

1. We ask that a manuscript source file is provided at Revision. Please upload your manuscript file as a .doc, .docx, .rtf or .tex.

 **Additional Editor Comments (if provided):****Reviewers' Comments:** Reviewer's Responses to Questions

**Comments to the Author**

1. Does this manuscript meet PLOS Digital Health’s publication criteria ? Is the manuscript technically sound, and do the data support the conclusions? The manuscript must describe methodologically and ethically rigorous research with conclusions that are appropriately drawn based on the data presented.

Reviewer #1: Yes

Reviewer #2: Yes

2. Has the statistical analysis been performed appropriately and rigorously?

Reviewer #1: Yes

Reviewer #2: Yes

3. Have the authors made all data underlying the findings in their manuscript fully available (please refer to the Data Availability Statement at the start of the manuscript PDF file)?

Reviewer #1: Yes

Reviewer #2: Yes

4. Is the manuscript presented in an intelligible fashion and written in standard English?

Reviewer #1: Yes

Reviewer #2: Yes

5. Review Comments to the Author

Reviewer #1: Significant Comments

- Part of the background discusses how a shorter interval EEG would be more feasible for resourse poor countries. These deep learning technologies and the EEG data used for the study don’t seem feasible in these settings. How do you expect that this can translate to these populations.

- Methods seem sound for training and test sets

- Excellent description of SPH and SOP goals.

- ? Wonder if it would be worth doing a subgroup analysis between the two datasets to see if that changes the results - only 2 (4%) with seizures in the cork group.

- ? How does this short horizon prediction equate to long term seizure risk?

- Some concerns about the poor generalizability from the per-subject analyses

- Appropriate discussion of limitations of the study design, results, generalizability

Minor Comments

- Would be helpful to have a brief background on who is at risk for these neonatal seizures/encephalopathies - briefly touched on in methodes, etc.

Reviewer #2: 1. The paper is well written and very detailed, with much of the information found in the supplementary materials. However, I suggest moving some of the tables into the main text for better readability. In particular, the table comparing different methods would be more useful in the main body.

2. The 3–7 minute lead time is a major strength because it directly addresses the need for quicker seizure warnings. However, it would help to clearly explain which specific clinical decisions or interventions could be enabled by this short warning window, and how these differ from those triggered by longer-horizon methods.

3. The authors tested multiple models (ConvLSTM, ResNet, Transformer, etc.) but focused on ConvLSTM. Please add more details on why ConvLSTM outperformed the others. It may help to note that ConvLSTM can capture both short-term features via CNN layers and longer temporal dependencies via LSTM layers.

4. The authors mentioned that only the top three QEEG feature categories were used. Please clarify why these three were selected and whether other features would have significantly changed the results.

5. More information is needed on how the authors shaped the data input for ConvLSTM. Please specify the dimensionality of the input and describe any normalization or regularization steps the authors applied.

6. The dataset appears to be imbalanced. Did the authors consider using weighted losses or focal losses to address this issue? Such methods can help prevent the model from overfitting to the majority class.

7. While the authors report the overall false detection rate, it would be helpful to discuss how these false alarms are distributed. For example, do certain subjects experience many more false alarms, or do they occur randomly?

8. It would be useful to add a brief discussion about how the authors' approach might be adapted or extended for continuous, real-time monitoring in a clinical setting. This could include how often the model updates its predictions or how it handles continuous data streams.

9. Currently, only the code for generating QEEG features is publicly available. Please consider sharing the full data processing and model training code to help other researchers reproduce the results. This would strengthen the paper’s overall impact and transparency.

6. PLOS authors have the option to publish the peer review history of their article (what does this mean? ). If published, this will include your full peer review and any attached files.

**Do you want your identity to be public for this peer review?** For information about this choice, including consent withdrawal, please see our Privacy Policy .

Reviewer #1: No

Reviewer #2: No

---

## [Decision Letter · Decision Letter 1]

Short-Horizon Neonatal Seizure Prediction Using EEG-Based Deep Learning

PDIG-D-25-00031R1

Dear Dr. Bernardo,

We are pleased to inform you that your manuscript 'Short-Horizon Neonatal Seizure Prediction Using EEG-Based Deep Learning' has been provisionally accepted for publication in PLOS Digital Health.

Best regards,

Hualou Liang

Academic Editor

PLOS Digital Health

**Additional Editor Comments (if provided):**

**Reviewer Comments (if any, and for reference):**

Reviewer's Responses to Questions

**Comments to the Author**

1. If the authors have adequately addressed your comments raised in a previous round of review and you feel that this manuscript is now acceptable for publication, you may indicate that here to bypass the “Comments to the Author” section, enter your conflict of interest statement in the “Confidential to Editor” section, and submit your "Accept" recommendation.

Reviewer #1: All comments have been addressed

2. Does this manuscript meet PLOS Digital Health’s publication criteria ? Is the manuscript technically sound, and do the data support the conclusions? The manuscript must describe methodologically and ethically rigorous research with conclusions that are appropriately drawn based on the data presented.

Reviewer #1: Yes

3. Has the statistical analysis been performed appropriately and rigorously?

Reviewer #1: Yes

4. Have the authors made all data underlying the findings in their manuscript fully available (please refer to the Data Availability Statement at the start of the manuscript PDF file)?

Reviewer #1: Yes

5. Is the manuscript presented in an intelligible fashion and written in standard English?

Reviewer #1: Yes

6. Review Comments to the Author

Reviewer #1: Thank you for allowing me to re-review this article. As before, the authors have done an excellent job. They have addressed all of my questions/comments as well as those from other reviewers that are included in this re-submission. I am impressed by the work that has been done in this project and hope o see more from this group in the future.

7. PLOS authors have the option to publish the peer review history of their article (what does this mean? ). If published, this will include your full peer review and any attached files.

**Do you want your identity to be public for this peer review?** For information about this choice, including consent withdrawal, please see our Privacy Policy .

Reviewer #1: No
